# A Parallel Corpus for Vietnamese Central-Northern Dialect Text Transfer

**Thang Le**
VinAI Research, Vietnam
`v.thangld16@vinai.io`

**Luu Anh Tuan***
Nanyang Technological University, Singapore
`anhtuan.luu@ntu.edu.sg`

## Abstract

The Vietnamese language embodies dialectal variants closely attached to the nation's three macro-regions: the Northern, Central and Southern regions. As the northern dialect forms the basis of the standard language, it's considered the prestige dialect. While the northern dialect differs from the remaining two in certain aspects, it almost shares an identical lexicon with the southern dialect, making the textual attributes nearly interchangeable. In contrast, the central dialect possesses a number of unique vocabularies and is less mutually intelligible to the standard dialect. Through preliminary experiments, we observe that current NLP models do not possess understandings of the Vietnamese central dialect text, which most likely originates from the lack of resources. To facilitate research on this domain, we introduce a new parallel corpus for Vietnamese central-northern dialect text transfer. Via exhaustive benchmarking, we discover monolingual language models' superiority over their multilingual counterparts on the dialect transfer task. We further demonstrate that fine-tuned transfer models can seamlessly improve the performance of existing NLP systems on the central dialect domain with dedicated results in translation and text-image retrieval tasks.

## 1 Introduction

Owing to the rapid development of natural language processing in the past few years, research on Vietnamese NLP have also benefited in terms of resources and modelling techniques. In particular, many task-specific Vietnamese datasets and dedicated language models have been released to the community (Dao et al., 2022a, Nguyen et al., 2018, Nguyen et al., 2022, Nguyen et al., 2020b, Nguyen and Nguyen, 2021, Tran et al., 2022a, Nguyen and Nguyen, 2020). These benchmarks and models, as they facilitate the research on Vietnamese com-

putational linguistics, have limitations in that they solely focus on the standard Vietnamese text.

| | Vietnamese Input Text | | Gold Translation |
|---|---|---|---|
| | răng tự nhiên ngá cực kỳ luôn [Central Dialect] | sao tự nhiên ngửa cực kỳ luôn [Northern Dialect] | |
| Google Translate | natural teeth are very yawn | Why is it so itchy all of a sudden? | Oh I feel so itchy |
| Yandex Translate | natural teeth are extremely toothed | why does it naturally itch extremely well | |
| ChatGPT | My natural teeth are extremely sharp | why does it suddenly itch so much all the time | |

Figure 1: Industry-level translation systems respond differently regarding the central dialect

Geographically, Vietnamese provinces are categorized into three macro-regions: the Northern, Central and Southern regions - bringing forth different dialects (Hip, 2009). Among these, the northern dialect is often treated as the standard i.e. the de-facto text style of the language (Phạm and Mcleod, 2016). Compared to the northern, the southern dialect differs in terms of pronunciation but is mostly similar in terms of lexicon, making the two mutually intelligible (Shimizu and Masaaki, 2021 , Son, 2018). In contrast, the central dialect, besides the phonology deviation, possesses a significant number of vocabulary differences and is thus less mutually intelligible to the remaining two (Michaud et al., 2015, Pham, 2019). In fact, the central dialect is often perceived as "peculiar" or even "difficult to understand" by speakers of other regions due to the existence of various unique words (Hip, 2009, Pham, 2019). With a population comprising of nearly one-third of the country, the central region, along with its dialect, remains an important and idiosyncratic part of the nation's culture (Handong et al., 2020, Phạm and Mcleod, 2016, Pham, 2005). However, existing research works, despite massive advancement in recent years, have mainly focused on the standard dialect so far (Truong et al.,

---

*Corresponding Author

2021, Nguyen et al., 2020a, Nguyen et al., 2017, Nguyen et al., 2020c, Lam et al., 2020), neglecting other variants with potentially rich linguistic values. Furthermore, state-of-the-art industry-level NLP systems, regardless of being production-ready, do not possess understandings of the non-standard dialect. An illustrative example is presented in Figure 1 where a text utterance in the central dialect is input into several renowned translation systems including Google Translate[1] & Yandex Translate[2]. Here the central-style utterance differs from the northern variety at two words *răng* and *ngá*, which corresponds to the words *sao* and *ngứa* in the northern counterpart. The word *răng* means *why/how* in the central dialect, but it also means *teeth* in the northern (standard) dialect. In contrast, the word *ngá* means *itchy* and is lexically unique to the central dialect. Taken together, the input text can be translated as *For some reasons I start to feel really itchy* or simply *Oh I feel so itchy*. We can observe that while the translation outputs of the northern-style utterance are highly relevant, the predicted outputs with respect to the central-style utterance are inapposite. In particular, all systems seem to mistake the meaning of the word *răng* as *teeth* and do not properly grasp the meaning of the word *ngá*. Despite a slight lexical difference between the two dialects, no system manages to correctly translate the central-style utterance, or even close, producing completely unconnected contents. This phenomenon can be considered a well-known bias in current NLP research, where developed models only function towards a particular, majority group of users, ignoring the needs of other minor communities e.g. translation for high-resource languages versus low-resource ones (Lignos et al., 2019, ter Hoeve et al., 2022, Ghosh and Caliskan, 2023). In the case of Vietnamese language, it manifested as the dialectal bias where the non-prestige (i.e. central) is not comprehended, even by well-tailored systems - a conceivable issue stemming from the lack of appropriate training data.

Inclined to provide a remedy to this dilemma, we introduce a new parallel corpus encompassing the central and northern dialect. Constructed through manual effort under strict quality-control, the corpus is specifically designed for the dialect transfer task, with meaningful applications towards facilitating the development of more well-rounded NLP

models that are not only potent towards the standard text but can also handle the central-style wordings. We extensively evaluate several monolingual and multilingual language models on their ability to shift the dialect of an input utterance, which requires deep linguistics understandings of the Vietnamese language. In addition, we provide experiments and discussions on the corpus's applications beyond mere dialect transfer, that is, adapting prevailing NLP models to the central dialect domain without the needs to re-train.

Our contributions can be summarized as follows:

- We introduce a new parallel corpus for central-northern dialect text transfer. We extensively benchmark the capacities of several monolingual and multilingual language models on the task, as well as their abilities to discern between the two dialects.

- We find that for the dialect transfer task, the monolingual models consistently and significantly outperform the multilingual models. In addition, we observe that all experimented generative models suffer from a fine-tuning - pre-training mismatch.

- We show that the competencies of existing models on downstream tasks, including translation and text-image retrieval, degrade when confronting central-style expressions. We further demonstrate that through fine-tuning dialect transfer adapters, the efficacies of these models in the central dialect domain can be tremendously improved without the needs to re-train.

## 2 Dataset Construction

In this section, we describe the procedure to construct a parallel corpus for Vietnamese central-northern dialect text transfer.

### 2.1 Procedure

Since there are subtle deviations among provinces in the same region, to improve the annotation consistency, we recruited central annotators whose hometowns are located in Ha Tinh - a representative province of the central region, and northern annotators who were born and grew up in Ha Noi - the country's capital and also a major metropolitan area of the northern region (Nguyen et al., 2006, Tran et al., 2022b). We also required each annotator to have familiarity with the other dialect (e.g.

---

[1]https://translate.google.com/
[2]https://translate.yandex.com/

through work, living experience, etc). Annotators first underwent a one-week training provided by a linguistics expert to assimilate the textual distinctions between the central and northern dialects. It's important that the annotators clearly grasp the dialectal differences in text styles. Before commencing the construction process, the annotators must pass an eligibility test where they are tasked with annotating 10 prototype samples [3]. Annotators whose annotation validity did not exceed 80% were re-trained and had to re-take the test until qualified. Eventually, 6 central annotators and 6 northern ones were recruited. We next describe the two steps employed to construct the parallel corpus. The first step only involves central annotators while the second requires the participation of all members.

**Step 1 - Central-style Corpus Creation.** There are no publicly available contents that are solely in the style of central dialect text. In order to construct a parallel corpus, we need to grasp a collection of raw central-style text utterances[4]. To this end, we first divided the central annotators into groups of two people. Each group was asked to act out pre-designed conversation scenarios where they chat with each other employing the central dialect. These scenarios were manually preset, ensuring that the communications are conducted in diverse situations (e.g. friends' casual chatting on romantic stories, workers planning an afterparty, etc), adding up to a number of 112 conversations in total. For every three rounds, we randomly swapped members between groups to maintain fresh perspectives. Upon completion, we asked every central annotator to pick out messages that are *central-style specific*. A message is considered *central-style specific* if it contains words with meanings, or lexical appearances, unique to the central dialect. Naturally communicating, not every message is *central-style specific*, as the central dialect also shares certain lexicon similarities with its northern counterpart. We observed that for every *central-style specific* message picked out, it received at least 4/6 votes from the annotators, indicating high-level uniformity. We selected all messages with full votes, and

held discussion sessions among annotators to resolve the messages with partial votes, in which the linguistics specialist also participated [5]. Ultimately, we obtained a set of 3761 *central-style specific* text utterances.

**Step 2 - Dialect Conversion.** For the obtained *central-style* corpus, we need to construct matching *northern-style* utterances. For this purpose, each central annotator was first paired with a distinct northern annotator. We then divided the raw samples into 10 folds, and evenly distributed them to each pair of annotators. In order to annotate a sample, a central annotator must first highlight dialect-specific words[6] present in the utterance and convey their meanings, as well as the utterance's, to the northern annotator, who then had to produce an equivalent utterance in the *northern-style* that fluently conveys the same contents, and is as close in nuances as possible. Following this step, we acquired a compilation of 3761 parallel central-northern utterance pairs.

## 2.2 Statistics

| Dialect | #Samples | #Avg. syll. | #Avg. word |
|---------|----------|-------------|------------|
| Central | 3761 | 10.88 | 10.35 |
| Northern | 3761 | 10.97 | 10.13 |

Table 1: Dataset statistics.

We present the corpus's base statistics in Table 1. The corpus was originally constructed in a syllable-separated manner, which is the natural appearance of Vietnamese text (Nguyen et al., 2018). However, for the Vietnamese language, space is also used to segment syllables of the same words (Dinh et al., 2008). For example, the text utterance *"Tôi là nghiên cứu sinh"* comprises of 5 separate syllables [*"Tôi"*, *"là"*, *"nghiên"*, *"cứu"*, *"sinh"*] that composite 3 words [*"Tôi"*, *"là"*, *"nghiên cứu sinh"*]. A majority of works employed the RDRsegmenter software (Nguyen et al., 2018) for automatic word segmentation (Dao et al., 2022a, Dao et al., 2022b, Nguyen and Nguyen, 2021, Truong et al., 2021, Nguyen et al., 2020a, Nguyen et al., 2017). As the tool was trained on the standard text, we first investigated its reliability in segmenting central-style variants. For this purpose, we executed the tool on central-style sequences and

---

[3]The test features **step 1-2** described in Section 2.1

[4]We also experimented with converting from the northern text samples to the central dialect. However, we chose the reverse direction as the annotators had difficulties selecting which elements to convert. Since the northern dialect is standard, every word/phrase present in the utterance has literally the same style, whereas with the central dialect, it's relatively easy to identify non-standard elements to conduct conversion.

[5]Partially voted messages constitute 2% of the total messages being picked out.

[6]We follow guidelines provided by Dinh et al., 2008 to deduce word boundaries.

randomly select a subset of 100 samples where at least 2 word-merging operations were performed. Manually evaluating the tool's precision, we find that it achieved 96.25% in precision for this particular subdivision. However, correctly segmented words were mostly standard words. Since we had annotated boundaries for central-style words, we next applied the software on all central-style sequences and calculated its recall rate as well as error rate[7] regarding central-style words. We found the recall rate to be extremely low (3.04%) which validated our hypothesis that the segmenter was not aware of central-style (non-standard) words and necessitated manual efforts for these specific words (which we did). In contrast, the error rate was fairly small (1.81%) which, adding up the high precision measured earlier, indicated that the software conducted segmentations conservatively i.e. avoiding words it did not know. These preliminary inspections showed that **RDRsegmenter's predictions are prudently reliable on standard words** but **its effectiveness on central-style words is nugatory**. Following, we executed the tools to obtain automatic boundaries for standard words[8]. This resulted in the word-segmented version of the corpus.

| Type | 1-gram | 2-gram | 3-gram | 4-gram | 5-gram |
|------|--------|--------|--------|--------|--------|
| Syll. | 42.17 | 64.34 | 77.68 | 86.04 | 91.34 |
| Word | 44.93 | 67.76 | 80.97 | 88.74 | 93.41 |

Table 2: Percentage of novel n-grams (%).

We report the percentage of novel n-grams in the central-style samples with respect to the northern counterparts. At the unigram level, the two dialects have a nearly 50% lexical distinction, bespeaking the uniqueness of the central dialect's vocabulary.

## 2.3 Quality Control

| | Fleiss. Kappa | Overall Agree. (%) | Avg. rating |
|---|---|---|---|
| Conversion | 0.7164 | 84.9 | 4.45 |
| C-Word Label | 0.6320 | 78.1 | 4.16 |

Table 3: Inter-annotator agreement.

To validate the dataset's quality, we randomly designated 100 pair of samples and requested each annotator duo (central & northern) to rate their

agreement with the conversion ranging from 1 to 5. For each central participant, we further asked he/she to appraise the annotation for central-style words and provide consensus scores on a similar scale. We report the Fleiss' Kappa scores (Fleiss, 1971) along with overall agreement in Table 3. Compared to the conversion task, we observed that the compromise on central-style word label was slightly lower. We later held a meeting with the annotators to investigate the cause and found that it emanated from ambiguities in determining word boundaries. Nevertheless, the statistics signified substantial inter-annotator agreement in each commisson.

## 3 Task Formulation

**Dialect Text Transfer.** Given a text utterance $x$ provided in the style of dialect $a$, we need to convert it to a targeted dialect $b$ while preserving the meaning of $x$.

Formally, denote $x = [x_0, x_1...x_n]$ as the sequence of input tokens and $y = [y_0, y_1...y_m]$ as the desired output tokens, we would like to model the conditional distribution $P(y|x)$. Training involves minimizing the negative log-likelihood $\mathcal{L} = -\sum_{i=1}^{t} log(P_\theta(y_t|y_{<t}, x))$ where $\theta$ represents the model's parameters.

In this work, we consider two settings: central-to-northern and northern-to-central. Tackling the prior means that we can readily adapt existing Vietnamese NLP models to handle the central dialect domain whereas the latter aids in synthesizing central-style data from existing standard corpus. Both directions have widespread applications and can facilitate building intelligent agents with more inclusive comprehension and capabilities.

## 4 Experiments & Discussions

**Experiment Settings.** We partitioned the dataset with 80%/10%/10% ratios to form the training/validation/test splits. For benchmarking the dialect transfer task, we fine-tuned a set of pre-trained generative language models: *mBART* (multilingual BART) (Liu et al., 2020), *BARTpho-syllable* (Vietnamese BART, operating on syllable-level data), *BARTpho-word* (similar to the prior but uses word-level data), *BARTpho-syllable-base* and *BARTpho-word-base* (the base variants) (Tran et al., 2022a). During training, we used a batch size of 32 and a learning rate of $1e-6$ along with the AdamW (Loshchilov and Hutter, 2017) optimizer with lin-

---

[7]This refers to the percentage of sequences where the automatically obtained central-style word boundaries violated the manually annotated ones

[8]We manually resolved cases where the tool violated annotations of central-style words (68/3761).

ear decay schedulers. All models were fine-tuned for a maximum of 300 epochs with early stopping. All settings were implemented with the PyTorch (Paszke et al., 2019) framework and the Transformers (Wolf et al., 2019) library. For each model, the top-5 checkpoints with lowest validation losses were selected and evaluated on the test set. We used greedy decoding in all experiments unless explicitly mentioned otherwise. We also considered SBERT-based (Reimers and Gurevych, 2019) retrieval baselines as lower bounds. In particular, we adopted two multilingual SBERT models (Reimers and Gurevych, 2020) denoted as *Retrieval-M1* and *Retrieval-M2* along with a publicly available Vietnamese SBERT model hereafter abbreviated as *Retrieval-Vi*. In each direction, we first encoded the input utterance and retrieved the sample with the closest semantic distance[9] in the target dialect from the training set. For details on pre-trained checkpoints, please see Appendix A.

To evaluate the quality of predicted sequences, we adopted a set of automatic evaluation metrics: ROUGE (Lin, 2004), BLEU (Papineni et al., 2002) and METEOR (Banerjee and Lavie, 2005). For the predictions of generative models, we also conducted human evaluations in which each participant was presented with the gold sequence and predicted outputs from 5 systems [10], conditioned on 100 random test samples. Predicted outputs were shuffled and the participants were not aware of the different models. Each participant then picked out the sequence that he/she thought was the most suitable conversion. For each direction, we employed 3 participants with approriate backgrounds (e.g. central-origin participants for northern-to-central transition and vice versa). Upon voting, we further held a meeting to resolve conflicts among raters where a linguistics specialist also partook in. For the two directions, we obtained corresponding Fleiss' Kappa scores (Fleiss, 1971) of 0.6447 and 0.6284 which implied substantial agreement.

**Dialect Transfer.** We present the results for two transfer directions in Table 4 and 5. For both settings, the retrieval baselines perform significantly worse than the generative (GEN) models which is to be presumed. Inspecting the retrieval and GEN models, the monolingual ones consistently and remarkably outperform the multilingual counterparts on every metric. In terms of human preference,

the monolingual outputs are also chosen more frequently[11]. For the northern-to-central direction, these gaps rise by a large margin. In particular, the BARTpho-syllable-base model outperforms the mBART model by 11 ROUGE-1 points and 22 BLEU scores. The mBART model is also least preferred by human. This shows that the dialect transfer task requires deep understandings of the Vietnamese language, making the monolingual models more fitting and accordingly perform much better than the multilingual ones.

**Mismatch between pre-training and fine-tuning.** Traditionally, these generative models were pre-trained on standard Vietnamese text (Tran et al., 2022a, Liu et al., 2020), conditioning them to produce Vietnamese text of the central dialect (which is non-standard) might induce an undesirable mismatch. Given an evaluation metric $P$, denote $P_{CN}$ (central-to-northern) and $P_{NC}$ (northern-to-central) as the model's performance in each direction. We define $\delta_P = P_{CN} - P_{NC}$ as the performance difference when conditioned to generate the northern-style text versus the central-style one. To scrutinize the mentioned phenomenon, we illustrate $\delta_{ROUGE-1}$ in Figure 2. It can be seen that performance drops manifest in every model type across different decoding beams. The mBART model has the largest drop of roughly 8 points, where each monolingual model exhibits a drop of around 1-2 points. This again betokens the inferiority of multilingual model in the dialect transfer task.

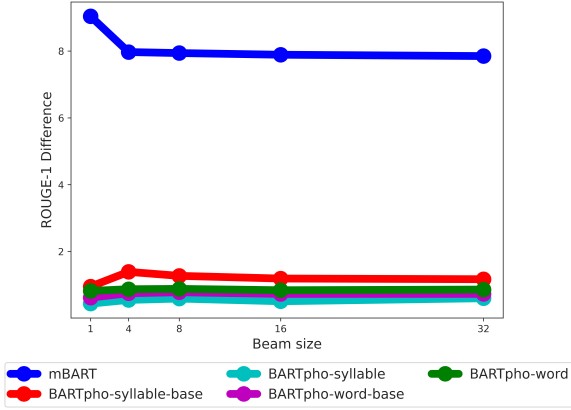

Figure 2: ROUGE-1 difference ($P_{CN} - P_{NC}$) with varying decoding beams.

**Zero-shot and Few-shot settings.** Large language models (LLMs) have been shown to possess

---

[9]We used cosine distance.

[10]In the event that two or more systems predict the same sequence, we de-duplicate the outputs to avoid biases

[11]As there exist cases where different models produced the same output and simultaneously got elected, the total number of votes are larger than the number of samples

| Model | R-1 | R-2 | R-L | BLEU | METEOR | Human |
|---|---|---|---|---|---|---|
| Retrieval-M1 | 20.90 ± 0.00 | 5.47 ± 0.00 | 18.10 ± 0.00 | 3.86 ± 0.00 | 14.58 ± 0.00 | - |
| Retrieval-M2 | 22.78 ± 0.00 | 6.26 ± 0.00 | 19.84 ± 0.00 | 4.01 ± 0.00 | 16.33 ± 0.00 | - |
| Retrieval-Vi | 24.77 ± 0.00 | 6.91 ± 0.00 | 21.55 ± 0.00 | 4.79 ± 0.00 | 18.12 ± 0.00 | - |
| mBART | 91.69 ± 0.05 | 85.58 ± 0.10 | 91.59 ± 0.05 | 84.42 ± 0.15 | 90.81 ± 0.08 | 19/100 |
| BARTpho-syllable-base | **94.38 ± 0.07** | **89.82 ± 0.09** | **94.34 ± 0.07** | 88.27 ± 0.12 | **93.75 ± 0.06** | 49/100 |
| BARTpho-syllable | 92.93 ± 0.08 | 87.34 ± 0.13 | 92.89 ± 0.09 | 85.90 ± 0.08 | 92.24 ± 0.09 | 20/100 |
| BARTpho-word-base | 93.75 ± 0.11 | 88.75 ± 0.15 | 93.73 ± 0.11 | 87.80 ± 0.19 | 93.19 ± 0.11 | 41/100 |
| BARTpho-word | 94.11 ± 0.12 | 89.34 ± 0.21 | 94.10 ± 0.12 | **88.38 ± 0.23** | 93.43 ± 0.12 | **55/100** |

Table 4: Results for central-to-northern dialect transfer (test set). We highlight the best results in each column and underline those in second.

| Model | R-1 | R-2 | R-L | BLEU | METEOR | Human |
|---|---|---|---|---|---|---|
| Retrieval-M1 | 17.18 ± 0.00 | 4.01 ± 0.00 | 14.90 ± 0.00 | 3.58 ± 0.00 | 11.30 ± 0.00 | - |
| Retrieval-M2 | 18.57 ± 0.00 | 4.81 ± 0.00 | 16.28 ± 0.00 | 3.78 ± 0.00 | 12.73 ± 0.00 | - |
| Retrieval-Vi | 24.41 ± 0.00 | 6.68 ± 0.00 | 19.99 ± 0.00 | 4.93 ± 0.00 | 17.80 ± 0.00 | - |
| mBART | 82.65 ± 0.53 | 72.95 ± 0.68 | 82.41 ± 0.53 | 64.64 ± 1.18 | 83.67 ± 0.40 | 6/100 |
| BARTpho-syllable-base | **93.43 ± 0.09** | **88.16 ± 0.16** | **93.40 ± 0.10** | **86.83 ± 0.17** | **92.92 ± 0.10** | **57/100** |
| BARTpho-syllable | 92.49 ± 0.13 | 86.40 ± 0.24 | 92.44 ± 0.13 | 84.97 ± 0.39 | 91.94 ± 0.17 | 40/100 |
| BARTpho-word-base | 93.13 ± 0.05 | 87.41 ± 0.14 | 93.08 ± 0.05 | 86.32 ± 0.11 | 92.41 ± 0.06 | **57/100** |
| BARTpho-word | 93.29 ± 0.04 | 87.59 ± 0.08 | 93.22 ± 0.04 | 86.26 ± 0.09 | 92.69 ± 0.05 | 51/100 |

Table 5: Results for northern-to-central dialect transfer (test set). We highlight the best results in each column and underline those in second.

emergent abilities that allow them to seamlessly generalize to unseen tasks (Brown et al., 2020). As a representative trial, we conducted experiments on ChatGPT[12][13](Ouyang et al., 2022, Ghosh and Caliskan, 2023) - a multilingual agent powered by state-of-the-art LLMs, for both the zero-shot and few-shot (5 exemplars) settings on the central-to-northern dialect transfer task. Since the chatbot itself possesses high expressiveness, we further asked it to provide explanations for the conducted operations. For each explanation point provided by the chatbot, we manually evaluated if it was valid (i.e. the explanation is linguistically correct, and the inferred operation also adds up). At the sequence level, we evaluated three aspects: fluency, style[14] and correctness (the predicted sequence must preserve the utterance's meaning and be fluent in the target dialect). We randomly selected 100 samples from the test set and also included the outputs of the *BARTpho-syllable-base* model as an upper bound (BARTpho for short). System outputs were anonymized and shuffled, each examined by 3 northern raters. As presented in Table 6, although ChatGPT could produce moderately stylized text, fewer than half of them were fluent and the correctness ratio was below 10%. The chatbot was capable of providing relatively well-grounded interpretation points with a nearly 30% valid ratio (Table 7), but its reliability still falls behind the fully supervised BARTpho model which achieved over 80% correctness ratio. We further observed that the few-shot setting helped improve the chatbot's performance, validating our initial hypothesis that the performance bottleneck in handling central-style inputs can be remedied with the provision of in-domain data. Nevertheless, with an observed overwhelming gap, multilingual large language models such as ChatGPT are far from replacing task-specific monolingual models such as BARTpho.

| | Correctness | Fluency | Style |
|---|---|---|---|
| ChatGPT (Zero-shot) | 5% | 37% | 54% |
| ChatGPT (Few-shot) | 9% | 39% | 58% |
| BARTpho (Fine-tuned) | 82% | 86% | 95% |

Table 6: Human analysis of ChatGPT and BARTpho's output qualities for central-to-northern dialect transfer.

**Adapt existing NLP models to the central dialect domain.** Re-training existing NLP models for the central-style inputs can be expensive, and thus building adapters to seamlessly adapt prevailing

---

[12]We use the version powered by GPT-3.5-Turbo.

[13]Responses collected during the period 28/05/2023 - 07/06/2023.

[14]We separately evaluated the style and fluency attributes as we noticed that certain predicted sequences did belong to the target dialect but were far from being fluent.

|  | Validity |
| --- | --- |
| ChatGPT (Zero-shot) | 27.34% (70/256) |
| ChatGPT (Few-shot) | 28.20% (75/266) |

Table 7: Human analysis of ChatGPT's interpretation points for central-to-northern dialect transfer.

models to the central-style text is a worth-exploring enactment. We first conduct experiments on the Vietnamese-English translation task with Google Translate as the base translation model[15]. We consider three types of input: the central-style utterance, the ground-truth northern-style utterance, and the one predicted with a *BARTpho-syllable-base* model. A total of 100 random samples were drawn from the test set, each rated by three participants[16] in terms of content and fluency on a 1-5 scale. The outputs were shuffled, and no participants were aware of the models' outputs beforehand. In the event that two or more outputs are identical, we conducted de-duplication to avoid biases. We report the average scores in Table 8. Here it is visible that the translation model degenerates substantially when confronting central-style inputs compared to the gold northern-style ones. In contrast, when these central-style inputs were transferred to the northern dialect (BARTpho), the translation qualities significantly improved and nearly matched that of the ground truth standards. These findings show that even though model's performance degrades when the input text belongs to the central (non-standard) dialect, we can construct adapters enabling existing NLP models to readily cope with the central-style inputs at high precision. We further present qualitative examples[17] in Figure 3.

| Text Style | Content | Fluency |
| --- | --- | --- |
| Northern (Gold) | 4.04 | 4.24 |
| Northern (BARTpho) | 3.96 | 4.21 |
| Central | 1.72 | 2.6 |

Table 8: Effects of dialect style on Vietnamese-English translation for Google Translate.

As an alternative application, we experimented with the text-image retrieval task. Specifically,

Figure 3: Dialect style affects Vietnamese-English translation.

we employ a multilingual CLIP model[18] (Radford et al., 2021, Reimers and Gurevych, 2020) to retrieve related images of a Vietnamese text query on the COCO dataset (Lin et al., 2014). We show a qualitative example on Figure 4 where the model receives input text both in the central dialect and the northern counterpart (transferred by the *BARTpho-syllable-base* model). The lower two lines depict images retrieved with the central-style query whereas the upper two lines contain those obtained with the converted standard query. We can see that the dialect of the input query largely affects the relevance of retrieved images especially when the query's key points stem from unique lexicons of the central (non-standard) dialect. In Figure 4, the central-style query contains the unique phrase *lấy gấy*[get married] which the CLIP model did not understand and thus retrieved irrelevant images (lower two lines). When processing the northern-style

[15]Responses collected during the period 28/05/2023 - 07/06/2023.

[16]All raters possess 7.0+ IELTS proficiencies

[17]Here the northern utterances were transferred by the *BARTpho-syllable-base* model.

[18]https://huggingface.co/sentence-transformers/clip-ViT-B-32-multilingual-v1

query, the model captured highly relevant outputs (upper two lines are all marriage-related images).

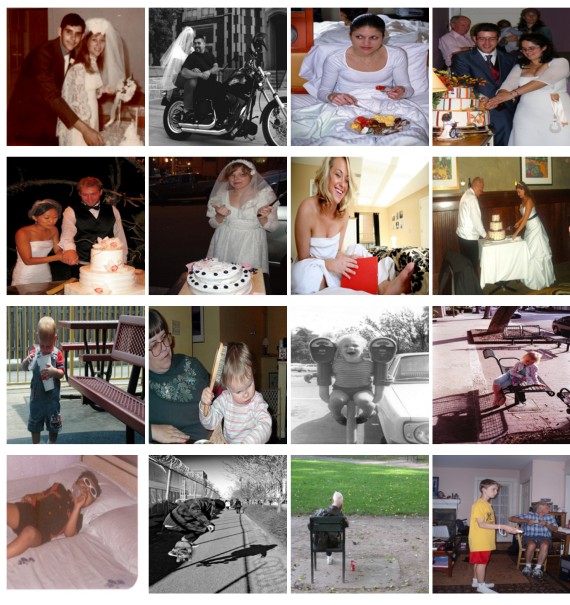

Northern-style: Tuổi mày thì lấy vợ được rồi chứ sao nữa
Central-style: Tuổi mi thì lấy gấy được rồi chi nựa
(English: You are at the age to get married)

Figure 4: Dialect style affects text-image retrieval.

**How discriminative is the central dialect ?** To study how anomalous the central dialect is on both the word- and sequence-level, we fine-tuned a set of auto-encoder language models on two tasks: central-style word extraction and dialect detection. The prior can be treated as a sequence labelling task whereas the latter resembles a sequence classification task. Note that the word extraction task is non-trivial, as central-style words can have the same lexical appearance with certain northern-style words in which cases the models have to disambiguate them through context information. For example, the word *răng* means *how/why* in the central dialect but also denotes *teeth* in the northern dialect. In our experiments, we employed the multilingual XLM-RoBERTa models (Conneau et al., 2019) and the monolingual PhoBERT models (Nguyen and Nguyen, 2020) (large and base variants)[19]. For every model, we picked the top-5 checkpoints with the best validation performance (accuracy or F1) and accordingly reported the mean and standard deviation. Training hyperparameters (learning rate, batch size, etc) remain the same as in previous experiments.

[19]PhoBERT-Base-V2 uses more pre-training data, but its architecture is the same as PhoBERT-Base

At the sequence-level (Table 9), it can be quite easy to distinguish between the two dialects where fine-tuned models achieved near-perfect accuracy. In contrast, at the word-level (Table 10), the central-style attributes are seemingly harder to catch. Nonetheless, we find that the detection (or extraction) performance is decent in general as the accuracy (or F1 score) of every model is well above 95%. We also observe that the monolingual PhoBERT models typically perform better than the XLM models, with the exception of the dialect detection task where the XLM-R-Large model outperforms the PhoBERT models on the test set (albeit by a small margin). On the central-style word extraction task, the PhoBERT models consistently outperform the XLM-R models on both the test and validation splits at the same architecture scale. We hypothesize that the dialect detection task only requires classifying the utterance's dialect on a sequence-level which can be relatively easy, whereas the word extraction task requires more extensive linguistics knowledge including locating word span and disambiguating central-style words from standard ones which the monolingual models do better at.

|  | Test Acc. | Val Acc. |
|---|---|---|
| PhoBERT-Large | 99.60 ± 0.00 | 99.73 ± 0.00 |
| XLM-RoBERTa-Large | 99.68 ± 0.07 | 99.87 ± 0.00 |
| PhoBERT-Base-V2 | 99.58 ± 0.06 | 99.87 ± 0.00 |
| PhoBERT-Base | 99.47 ± 0.00 | 99.89 ± 0.06 |
| XLM-RoBERTa-Base | 99.39 ± 0.12 | 99.60 ± 0.00 |

Table 9: Results on dialect detection.

|  | Test F1 | Val F1 |
|---|---|---|
| PhoBERT-Large | 97.03 ± 0.04 | 98.32 ± 0.02 |
| XLM-RoBERTa-Large | 96.89 ± 0.04 | 97.94 ± 0.05 |
| PhoBERT-Base-V2 | 97.21 ± 0.06 | 97.86 ± 0.04 |
| PhoBERT-Base | 96.77 ± 0.09 | 98.02 ± 0.02 |
| XLM-RoBERTa-Base | 96.62 ± 0.09 | 97.56 ± 0.02 |

Table 10: Results on central-style word extraction.

# 5 Related Works

Research on language varieties, specifically dialects, has been actively developed among many languages including those with Latin (Demszky et al., 2020, Kuparinen, 2023, Samardić et al., 2016, Kuparinen and Scherrer, 2023, Miletic and Scherrer, 2022, Dereza et al., 2023, Ramponi and Casula, 2023, Vaidya and Kane, 2023, Aji et al., 2022) and

non-Latin writing systems (Zaidan and Callison-Burch, 2011, Bouamor et al., 2018, Li et al., 2023). The research areas mostly revolve around dialect classification (Kanjirangat et al., 2023, Kuzman et al., 2023) and normalization (Kuparinen and Scherrer, 2023 , Hämäläinen et al., 2022), but also span related downstream tasks such as hate speech detection (Castillo-López et al., 2023), sentiment analysis (Srivastava and Chiang, 2023), part-of-speech tagging (Mæhlum et al., 2022) and eye-tracking (Li et al., 2023). For the Vietnamese language, dialect-related research remains bounded in traditional linguistics research (Pham, 2019, Hip, 2009, Tsukada and Nguyen, 2008, Son, 2018), whereas computational linguistics mainly puts the focus on the standard dialect (Nguyen et al., 2020b, Nguyen et al., 2020c, Lam et al., 2020). Amid the small number of works considering dialectal differences, only signal processing and speech-related tasks are explored (Hung et al., 2016a, Hung et al., 2016b, Schweitzer and Vu, 2016), centering on the tonal (phonetic) deviations between dialects while the textual attributes and their effects on downstream tasks remain unexplored. In recent years, a plethora of Vietnamese NLP datasets have been released to facilitate the development of downstream tasks including intent detection and slot filling (Dao et al., 2022b), speech translation (Nguyen et al., 2022), machine translation (Doan et al., 2021), named entity recognition (Truong et al., 2021) and text-to-sql (Nguyen et al., 2020a). These corpora have the same limitations in that they do not take into account the (non-standard) central dialect and only target the standard text. While they accelerate the progress on a number of tasks, systems trained on these datasets potentially carry the same drawbacks in that they are not apt to confront the central-style wordings which might in turn cause the existing unfairness to become more severe.

As growing needs for text style transfer emerge, the field has been actively receiving attention from the research communities (Jin et al., 2020). The settings vastly differ per situational basis, ranging from business use cases such as formality (Briakou et al., 2021), politeness (Madaan et al., 2020), authorship (Carlson et al., 2017), simplicity (den Bercken et al., 2019) to scenarios that aim at mitigating social issues such as toxicity (dos Santos et al., 2018), sarcasm (Tay et al., 2018b), gender (Prabhumoye et al., 2018), sentiment (He and McAuley, 2016; Tay et al., 2018b,a,c), biases

(Voigt et al., 2018). Among them, many are developed with extended applications in mind i.e. facilitating the progress of other tasks such as paraphrasing (Yamshchikov et al., 2020), summarizing (Bertsch et al., 2022) or producing style-specific translation (Wu et al., 2020). In our work, we focus on a more language-oriented setting, that is, the transfer between different dialects (i.e. the central and northern dialects), with meaningful pertinence towards more inclusive NLP models.

# 6 Conclusion

In this paper, we tackle the dialect transfer problem for the Vietnamese language. In particular, we introduce a new benchmark for Vietnamese central-northern dialect text transfer. Through immense experiments, we discover the deficiencies of multilingual models for the task compared to the monolingual counterparts. We also highlight the performance bias of existing NLP systems regarding the Vietnamese central dialect. As a prospective remedy, we further demonstrate that the fine-tuned transfer modules can empower existing models to address the central-style wordings without the needs for re-training.

# 7 Limitations

Although our work addresses practical problems specific to the Vietnamese language and the central dialect, the corpus was constructed with the participation of annotators from representative provinces only (i.e. Ha Tinh and Ha Noi). While this decision consolidates the annotation consistencies, the corpus only represents a major portion of the idiosyncratic features of the two dialects, and not their entirety. Therefore, closer inspection at the subtle deviations between other related provinces can provide more insightful looks into the characteristics of the two dialects. Additionally, although we put more focus on the applications of central-to-northern transfer in this work as they aid in readily adapting existing NLP models to the central-style text, the reverse direction also appeals as it can facilitate the synthesis of central-style task-specific data for Vietnamese NLP research. As the experiments demonstrate a decent level of performance in fine-tuned models, leveraging them to synthesize and assemble targeted resources in the central dialect can be an impactful direction for future works.

## Acknowledgement

We thank the anonymous reviewers for their constructive feedback and helpful suggestions.

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

## A    Pre-trained Checkpoints

The list of pre-trained models can be found in Table 11.

## B    Additional Results

We present the detailed results obtained with different beam widths in Table 12 and 13. For each model, we highlight the beam size that brings about the highest score. On the central-to-northern direction, most models obtain slight performance gains when expanding beam width, except for *BARTpho-word*. On the reverse direction, only the *mBART* model benefits from increasing beam size, whereas the greedy decoding suffices for all monolingual *BARTpho* models and raising beam sizes only triggers degradation due to the curse of beam search. This indicates that the optimum learnt by the monolingual models are closer to the global optimum and thus making greedy decoding an ideal selection, whereas the optimum learnt by the *mBART* model diverges from the global optimum and therefore increasing the beam width helps in producing better outputs.

| Model | Namespace |
|-------|-----------|
| Retrieval-M1 | sentence-transformers/paraphrase-multilingual-MiniLM-L12-v2 |
| Retrieval-M2 | sentence-transformers/paraphrase-multilingual-mpnet-base-v2 |
| Retrieval-Vi | keepitreal/vietnamese-sbert |
| mBART | facebook/mBART-large-cc25 |
| BARTpho-syllable-base | vinai/BARTpho-syllable-base |
| BARTpho-syllable | vinai/BARTpho-syllable |
| BARTpho-word-base | vinai/BARTpho-word-base |
| BARTpho-word | vinai/BARTpho-word |
| XLM-RoBERTa-Base | xlm-roberta-base |
| XLM-RoBERTa-Large | xlm-roberta-large |
| PhoBERT-Base | vinai/phobert-base |
| PhoBERT-Base-V2 | vinai/phobert-base-v2 |
| PhoBERT-Large | vinai/phobert-large |
| Multilingual CLIP | sentence-transformers/clip-ViT-B-32-multilingual-v1 |

Table 11: List of pre-trained models and their according namespaces in Huggingface.

| | R-1 | R-2 | R-L | BLEU | METEOR |
|---|---|---|---|---|---|
| **mBART** | | | | | |
| Beam 1 (Greedy) | 91.69 ± 0.05 | 85.58 ± 0.10 | 91.59 ± 0.05 | 84.42 ± 0.15 | 90.81 ± 0.08 |
| Beam 4 | 91.81 ± 0.09 | 85.77 ± 0.12 | 91.71 ± 0.09 | 84.59 ± 0.13 | 90.99 ± 0.15 |
| **Beam 8** | **91.82 ± 0.07** | **85.78 ± 0.08** | **91.72 ± 0.07** | **84.62 ± 0.10** | **91.00 ± 0.08** |
| Beam 16 | 91.78 ± 0.06 | 85.75 ± 0.08 | 91.68 ± 0.06 | 84.57 ± 0.14 | 90.97 ± 0.06 |
| Beam 32 | 91.74 ± 0.05 | 85.66 ± 0.07 | 91.65 ± 0.05 | 84.52 ± 0.11 | 90.88 ± 0.07 |
| **BARTpho-syllable-base** | | | | | |
| Beam 1 (Greedy) | 94.38 ± 0.07 | 89.82 ± 0.09 | 94.34 ± 0.07 | 88.27 ± 0.12 | 93.75 ± 0.06 |
| Beam 4 | 94.46 ± 0.06 | 89.84 ± 0.08 | 94.42 ± 0.06 | 88.32 ± 0.13 | 93.80 ± 0.05 |
| Beam 8 | 94.47 ± 0.05 | 89.85 ± 0.06 | 94.43 ± 0.05 | 88.21 ± 0.10 | 93.83 ± 0.03 |
| Beam 16 | 94.47 ± 0.05 | 89.85 ± 0.08 | 94.43 ± 0.05 | 88.17 ± 0.13 | 93.81 ± 0.05 |
| **Beam 32** | **94.49 ± 0.05** | **89.90 ± 0.08** | **94.45 ± 0.05** | **88.23 ± 0.13** | **93.82 ± 0.05** |
| **BARTpho-syllable** | | | | | |
| Beam 1 (Greedy) | 92.93 ± 0.08 | 87.34 ± 0.13 | 92.89 ± 0.09 | 85.90 ± 0.08 | 92.24 ± 0.09 |
| Beam 4 | 93.00 ± 0.06 | 87.33 ± 0.13 | 92.95 ± 0.07 | 85.77 ± 0.18 | 92.24 ± 0.07 |
| Beam 8 | 93.00 ± 0.05 | 87.35 ± 0.08 | 92.95 ± 0.05 | 85.84 ± 0.13 | 92.27 ± 0.05 |
| Beam 16 | 92.93 ± 0.09 | 87.24 ± 0.14 | 92.88 ± 0.09 | 85.69 ± 0.22 | 92.20 ± 0.10 |
| **Beam 32** | **93.01 ± 0.07** | **87.39 ± 0.09** | **92.96 ± 0.07** | **85.84 ± 0.17** | **92.28 ± 0.08** |
| **BARTpho-word-base** | | | | | |
| Beam 1 (Greedy) | 93.75 ± 0.11 | 88.75 ± 0.15 | 93.73 ± 0.11 | 87.80 ± 0.19 | 93.19 ± 0.11 |
| Beam 4 | 93.79 ± 0.14 | 88.95 ± 0.19 | 93.78 ± 0.14 | 88.01 ± 0.20 | 93.27 ± 0.15 |
| **Beam 8** | **93.80 ± 0.07** | **88.95 ± 0.10** | **93.78 ± 0.07** | **88.02 ± 0.16** | **93.27 ± 0.07** |
| Beam 16 | 93.78 ± 0.08 | 88.93 ± 0.11 | 93.76 ± 0.08 | 88.00 ± 0.17 | 93.25 ± 0.09 |
| Beam 32 | 93.78 ± 0.12 | 88.93 ± 0.16 | 93.76 ± 0.12 | 88.00 ± 0.19 | 93.25 ± 0.13 |
| **BARTpho-word** | | | | | |
| **Beam 1 (Greedy)** | **94.11 ± 0.12** | **89.34 ± 0.21** | **94.10 ± 0.12** | **88.38 ± 0.23** | **93.43 ± 0.12** |
| Beam 4 | 94.09 ± 0.14 | 89.30 ± 0.25 | 94.07 ± 0.14 | 88.32 ± 0.28 | 93.41 ± 0.15 |
| Beam 8 | 94.09 ± 0.15 | 89.30 ± 0.27 | 94.07 ± 0.15 | 88.32 ± 0.28 | 93.41 ± 0.17 |
| Beam 16 | 94.05 ± 0.11 | 89.26 ± 0.21 | 94.04 ± 0.11 | 88.25 ± 0.20 | 93.37 ± 0.12 |
| Beam 32 | 94.07 ± 0.14 | 89.27 ± 0.25 | 94.05 ± 0.14 | 88.29 ± 0.23 | 93.38 ± 0.16 |

Table 12: Effects of different beam sizes (central-to-northern).

|  | R-1 | R-2 | R-L | BLEU | METEOR |
|---|---|---|---|---|---|
| **mBART** | | | | | |
| Beam 1 (Greedy) | 82.65 ± 0.53 | 72.95 ± 0.68 | 82.41 ± 0.53 | 64.64 ± 1.18 | 83.67 ± 0.40 |
| Beam 4 | 83.84 ± 0.54 | 74.23 ± 0.56 | 83.67 ± 0.51 | 66.92 ± 1.12 | 84.45 ± 0.42 |
| Beam 8 | 83.88 ± 0.52 | 74.28 ± 0.62 | 83.70 ± 0.52 | 66.94 ± 1.09 | 84.53 ± 0.45 |
| Beam 16 | 83.89 ± 0.47 | 74.28 ± 0.60 | 83.68 ± 0.50 | 66.82 ± 1.04 | 84.61 ± 0.41 |
| **Beam 32** | **83.89 ± 0.51** | **74.35 ± 0.64** | **83.69 ± 0.53** | **66.88 ± 1.13** | **84.61 ± 0.42** |
| BARTpho-syllable-base | | | | | |
| **Beam 1 (Greedy)** | **93.43 ± 0.09** | **88.16 ± 0.16** | **93.40 ± 0.10** | **86.83 ± 0.17** | **92.92 ± 0.10** |
| Beam 4 | 93.07 ± 0.08 | 87.49 ± 0.20 | 93.04 ± 0.09 | 86.15 ± 0.17 | 92.44 ± 0.07 |
| Beam 8 | 93.20 ± 0.12 | 87.76 ± 0.23 | 93.17 ± 0.12 | 86.48 ± 0.26 | 92.55 ± 0.12 |
| Beam 16 | 93.28 ± 0.13 | 87.88 ± 0.25 | 93.25 ± 0.14 | 86.57 ± 0.22 | 92.64 ± 0.12 |
| Beam 32 | 93.32 ± 0.16 | 87.94 ± 0.30 | 93.29 ± 0.17 | 86.61 ± 0.27 | 92.72 ± 0.20 |
| BARTpho-syllable | | | | | |
| **Beam 1 (Greedy)** | **92.49 ± 0.13** | **86.40 ± 0.24** | **92.44 ± 0.13** | **84.97 ± 0.39** | **91.94 ± 0.17** |
| Beam 4 | 92.45 ± 0.06 | 86.30 ± 0.10 | 92.40 ± 0.05 | 84.69 ± 0.18 | 91.88 ± 0.09 |
| Beam 8 | 92.41 ± 0.13 | 86.23 ± 0.23 | 92.36 ± 0.12 | 84.56 ± 0.31 | 91.83 ± 0.16 |
| Beam 16 | 92.42 ± 0.13 | 86.22 ± 0.23 | 92.38 ± 0.13 | 84.61 ± 0.31 | 91.85 ± 0.15 |
| Beam 32 | 92.41 ± 0.11 | 86.21 ± 0.23 | 92.36 ± 0.12 | 84.55 ± 0.34 | 91.83 ± 0.15 |
| BARTpho-word-base | | | | | |
| **Beam 1 (Greedy)** | **93.13 ± 0.05** | **87.41 ± 0.14** | **93.08 ± 0.05** | **86.32 ± 0.11** | **92.41 ± 0.06** |
| Beam 4 | 93.04 ± 0.04 | 87.37 ± 0.13 | 92.99 ± 0.04 | 86.13 ± 0.13 | 92.31 ± 0.05 |
| Beam 8 | 93.02 ± 0.07 | 87.31 ± 0.17 | 92.97 ± 0.07 | 86.13 ± 0.15 | 92.31 ± 0.08 |
| Beam 16 | 93.05 ± 0.05 | 87.34 ± 0.15 | 93.00 ± 0.05 | 86.17 ± 0.13 | 92.34 ± 0.06 |
| Beam 32 | 93.05 ± 0.05 | 87.34 ± 0.15 | 92.99 ± 0.05 | 86.16 ± 0.13 | 92.34 ± 0.06 |
| BARTpho-word | | | | | |
| **Beam 1 (Greedy)** | **93.29 ± 0.04** | **87.59 ± 0.08** | **93.22 ± 0.04** | **86.26 ± 0.09** | **92.69 ± 0.05** |
| Beam 4 | 93.22 ± 0.06 | 87.51 ± 0.13 | 93.15 ± 0.06 | 86.24 ± 0.14 | 92.61 ± 0.07 |
| Beam 8 | 93.21 ± 0.05 | 87.51 ± 0.13 | 93.14 ± 0.05 | 86.24 ± 0.14 | 92.61 ± 0.07 |
| Beam 16 | 93.21 ± 0.05 | 87.51 ± 0.12 | 93.14 ± 0.05 | 86.26 ± 0.13 | 92.61 ± 0.07 |
| Beam 32 | 93.21 ± 0.06 | 87.50 ± 0.13 | 93.14 ± 0.06 | 86.24 ± 0.14 | 92.61 ± 0.07 |

Table 13: Effects of different beam sizes (northern-to-central).

## C  Attention Visualization

To better understand the generative process of the trained models, we visualize the last decoder layer's cross-attention maps of random test samples with BERTViz (Vig, 2019). Here we pick the central-to-northern transfer direction and choose the outputs of three models: *BARTpho-syllable-base*, *BARTpho-word-base* and mBART. In Figure 5, the token *đồ* (central-style) corresponds to the tokens *các kiểu* (northern-style). We can observe that the attention maps of the BARTpho models are more accurate, aligning with the target tokens, whereas the mBART model's attention maps are more vague, and not quite correct. In Figure 6, the token *ngài* (central-style) corresponds to the token *người* in the northern dialect. The attention maps of the three models are however, a bit off, with main focus on the preceding token *bốn* instead of

the token *người*. Still, we can perceive that in the cases of BARTpho models, there are more attention heads pointing to the token *người* than the mBART model.

## D  Prompts

For the zero-shot experiments with ChatGPT, we used the following template:

```
Convert the following Vietnamese central
dialect text utterance into the northern
dialect. Explain the difference and how
you do it.
Central Text: {Central-style test input}
Northern Text:
```

For the few-shot experiments, we randomly sample 5 exemplars from the training set for each prompt and used a similar template:

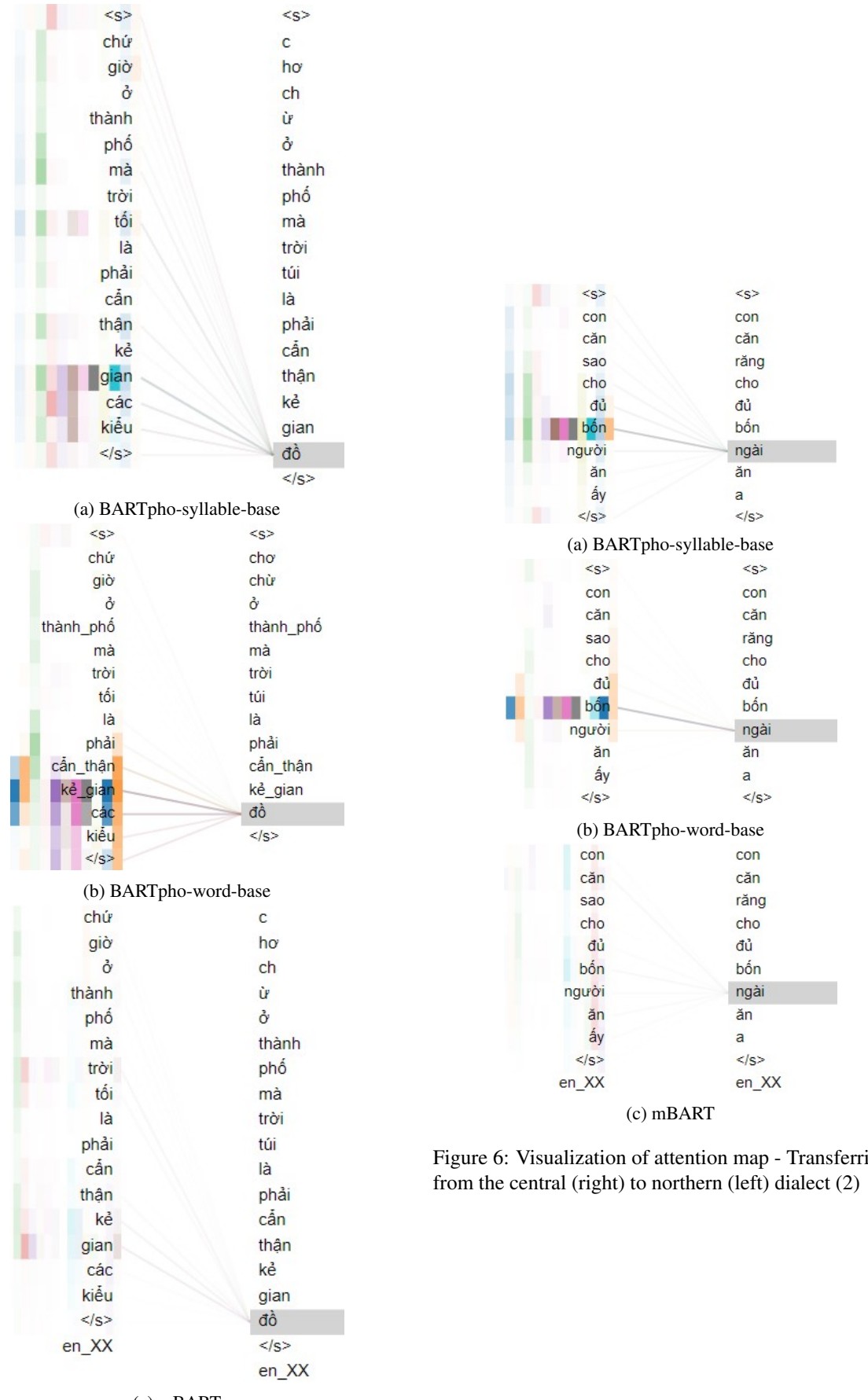

(a) BARTpho-syllable-base

(b) BARTpho-word-base

(c) mBART

Figure 5: Visualization of attention map - Transferring from the central (right) to northern (left) dialect (1)

(a) BARTpho-syllable-base

(b) BARTpho-word-base

(c) mBART

Figure 6: Visualization of attention map - Transferring from the central (right) to northern (left) dialect (2)

```
Here are a few examples of parallel
Vietnamese central-northern dialect text
utterance pairs.

Central Text: {Central-style input 1}
Northern Text: {Northern-style output 1}

Central Text: {Central-style input 2}
Northern Text: {Northern-style output 2}

...

Central Text: {Central-style input 5}
Northern Text: {Northern-style output 5}

Convert the following Vietnamese central
dialect text utterance into the northern
dialect. Explain the difference and how
you do it.
Central Text: {Central-style test input}
Northern Text:
```

Queries were performed via Poe's interface[20]. We observed that although the chatbot provided well-structured answers, the outputs were not always consistent (i.e. the explanations might differ from the predicted sequence), and were prone to hallucinations. We give an example of this phenomenon in Figure 7. In this example, the chatbot explains that it will change the word *mi* to *tôi*, but in the predicted output, the word *mi* is simply omitted instead of being replaced by the word *tôi*. Taken together, the predicted sequence is also wrong as the ground-truth sequence (northern-style) should be: "*thôi mày đừng nói linh tinh nữa là được còn gì*" (*It will be fine as long as you stop talking nonsense*), while the model's predicted output: "*thôi đừng nói dài dòng thêm nữa là được chưa*" means *Let's not talk about it any longer, okay?* (it's also an inarticulate northern utterance).

## E Examples

We present a few examples from the corpus in Table 14.

Figure 7: An example of ChatGPT's response (zero-shot). Here the explanations deviate from the predicted output.

| |
|---|
| **Central:** Quê mi nuôi tru có con mô to như con ni không ?
**Northern:** Quê mày nuôi trâu có con nào to như con này không ?
(*Is there any buffalo as big as this one in your hometown ?*) |
| **Central:** Gơ mi mần chi liều rứa !?
**Northern:** Ủa mày làm gì mà liều vậy !?
(*Why are you so reckless !?*) |
| **Central:** Tau ngồi cả buổi nỏ nghịch chi hết trơn
**Northern:** Tao ngồi cả buổi không nghịch gì hết luôn
(*I stayed quiet the whole time*) |

Table 14: Examples from the corpus.

[20]https://poe.com/ChatGPT