# OpenReview forum: "A Parallel Corpus for Vietnamese Central-Northern Dialect Text Transfer"
_EMNLP/2023/Conference — EMNLP 2023 Findings_

### Official Review · Reviewer_sxaD · 2023-08-02

**Soundness:** 3

**Excitement:**

4: Strong: This paper deepens the understanding of some phenomenon or lowers the barriers to an existing research direction.

**Missing References:**

- There has been a lot more work on Arabic since Zaidan and Callison-Burch, 2011. For example, the MADAR corpus: https://aclanthology.org/L18-1535.pdf
- Even though it is not open source (but only $25), the LDC has a corpus of Vietnamese that has multiple dialects, including the central dialect: https://catalog.ldc.upenn.edu/LDC2017S01 If the authors can not access it to explore and compare to it, I believe it is worth mentioning as an existing corpus.

**Paper Topic And Main Contributions:**

In this paper, the authors tackle the issue of dialectal representation in NLP for the Vietnamese language. The dialect of focus in this paper is central Vietnamese since it is more distinct than the "standard" northern dialect compared to the southern dialect. The main contribution of this paper is creating a parallel corpus of northern-central Vietnamese, with dialect transfer being the main purpose of this corpus. Other contributions include benchmarking "dialect transfer" using pretrained BART models and using them as "adaptors" for downstream NLG tasks.

**Questions For The Authors:**

## Questions
- QA: Line 085, the authors use commercial translators to showcase examples; in what day or version did they use them? Recording this information for future reference is important as these services often change (to a better or worse version).
- QB: It is unclear whether the dataset for the central Vietnamese dialect was manually annotated for word boundaries (segmentation by merging syllables). Related to this question, Table 2 is not clear. What does "both" mean in this context? And I am confused by what mistakes mean; I could not comprehend that from the text either.
- QC: The section about mismatch between pretraining and finetuning is not clear. Either I am missing something or the experiment/task is not well presented, so understanding Figure 2 is hard.

## Comments
- CA: BART is not an autoregressive model. It is more of a seq-2-seq model.
- CB: The captions under all the tables and figures are very brief, and they barely convey anything. For example, it is not clear in Tables and 5 and 6 it is not clear whether the reported numbers are on dev or test. Or what do the underlined numbers exactly mean. One can only guess, and that is not very helpful.

**Reasons To Accept:**

- A new dataset for a language variety, central Vietnamese, not present in the NLP scene. This is always welcomed, as it will enable more research to be conducted on central Vietnamese.
- A thorough process of creating the dataset and benchmarking experiments.
- Highlighting the need for dialect specific data for NLP research and that multilingual models are insufficient.
- The findings of annotation decisions and some experimentation results align with previous work in dialectal research, specifically Arabic.

**Reasons To Reject:**

- Sometimes, it is hard to follow the discussion of the results, especially since the table captions are very brief.
- In the contributions, the authors mention text-image retrieval, which is not discussed anywhere in the main body of the paper, only in the appendix. Anything discussed in the main body of the paper should be presented there.
- The terminology of some know models and approaches are misleading (see comment).

**Reproducibility:**

4: Could mostly reproduce the results, but there may be some variation because of sample variance or minor variations in their interpretation of the protocol or method.

**Reviewer Confidence:**

3: Pretty sure, but there's a chance I missed something. Although I have a good feel for this area in general, I did not carefully check the paper's details, e.g., the math, experimental design, or novelty.

**Typos Grammar Style And Presentation Improvements:**

- Any mention of Table and Figure should be capitalized.
- Line 082 Table A --> Figure 1
- There is plenty of space in the paper, so it is worth making Tables 5 and 6 larger.
- Line 072 exits --> existing
- Line 103 dialect --> dialects
- Adding examples from the corpus would be helpful.

---

> ### Author Rebuttal · Authors · 2023-08-28
>
> Dear reviewer,
>
> We greatly appreciate your detailed and constructive comments and suggestions. Below we address the main concerns raised in your review.
>
> **Question from the reviewer**
>
> > *Line 085, the authors use commercial translators to showcase examples; in what day or version did they use them? Recording this information for future reference is important as these services often change (to a better or worse version).*
>
> Responses from commercial translators were collected during the period from May 28th - June 7th. In addition, responses from the ChatGPT system were also gathered during this time. We thank the reviewer for the reminder. We will update these details in the paper.
>
> > *It is unclear whether the dataset for the central Vietnamese dialect was manually annotated for word boundaries (segmentation by merging syllables). Related to this question, Table 2 is not clear. What does "both" mean in this context? And I am confused by what mistakes mean; I could not comprehend that from the text either.*
>
> We apologize for the lack of clearness. The corpus contains pairs of text sequences, each belonging to the central and northern dialect, respectively.
>
> For the text sequences belonging to the central dialect, we manually annotated word boundaries for words specifically unique to this dialect as we wanted to examine its distinctive properties. However, there were standard words mixed in these sequences as well (since the two dialects shared the lexicon to a certain degree). Thus, we conducted trials to see if these words could be segmented with a public software [1]. In particular, we fed the whole central-style inputs to the software and verified the obtained word boundaries. If the tool’s precision were sufficiently high, it would indicate that the predictions were reliable (i.e. the word boundaries contained few mistakes). To confirm this, we randomly sampled 100 segmented sequences where at least two word-merging operations were conducted and calculated the tool’s precision accordingly. The results correspond to line 2 of Table 2, and the “both” here indicates that the precision score was calculated regarding both the central-style and standard words present in the inputs. As we observed that the precision was fairly high (96.2%), it indicated that the tool’s outputs contained few errors.
>
> Next, we examined the tool’s behavior with respect to the central-style words. Since we had manually annotated boundaries for these words, we could directly evaluate the automatic obtained spans based on the manually annotated ones. We use two criteria: recall (the percentage of groundtruth boundaries discovered by the tool) and mistakes (the percentage of sequences where the tool violated the groundtruth boundaries with wrong predictions, e.g. the groundtruth span could be <1,2,3> but the tool’s output was <2,3,4>). The results were presented in line 3 and 4 of Table 2. (Central) indicated that we only considered central-style words here, and (All) indicated that we evaluated on all central-style sequences. We noticed that the recall was low (3.04%) which supported our suspects that the tool could not detect central-style words properly. Meanwhile, the mistake ratio was small (1.81%), indicating that the tool was quite conservative and refrained from making uncertain decisions (i.e. those involving central-style words) which was a favorable characteristic.
>
> Ultimately, for the text sequences belonging to the central dialect, the central-style words were manually annotated whereas the other (standard) words were automatically segmented.
>
> For the text sequences which belong to the northern dialect (a.k.a standard text), we can safely apply the public software [1] to automatically obtain word boundaries as they have been properly benchmarked on the standard input domain.
>
> [1] Dat Quoc Nguyen, Dai Quoc Nguyen, Thanh Vu, Mark Dras, Mark Johnson: A Fast and Accurate Vietnamese Word Segmenter. LREC 2018
>
> > *The section about mismatch between pretraining and finetuning is not clear. Either I am missing something or the experiment/task is not well presented, so understanding Figure 2 is hard.*
>
> We apologize for the lack of clearness. We experimented with two transfer directions:  northern-to-central (NC) and central-to-northern (CN). Given an evaluation metric $P$, denote $P_{NC}$ and $P_{CN}$ as the model’s performance in each direction accordingly. We computed $\delta_P=P_{NC}-P_{CN}$ as the difference in model’s performance when conditioned to generate the central-style text versus the northern-style text. In our experiments, we observed that $\delta_P$ was huge across different models and (decoding) beam sizes. As an illustration, we visualized $\delta_{ROUGE-1}$ of each model in Figure 2 with varying (decoding) beam sizes. It could be inferred that all models faced increasing difficulties in generating central-style texts i.e. struggled in the northern-to-central direction, and it was especially the case for the mBART model.
>
> We linked this phenomenon to the pre-training phase of these models and conjectured that this was due to a pre-training and fine-tuning mismatch. In particular, the BARTpho model was pre-trained with denoising objectives on the Vietnamese Wikipedia and a news corpus [2], all of which were standard texts. The mBART model also underwent similar pre-training on standard texts but with higher volume (i.e. more data). As a result, these models had a higher tendency to generate (northern) standard texts rather than central-style texts, which caused the high values of $\delta_{ROUGE-1}$ as shown in Figure 2.
>
> [2] https://github.com/binhvq/news-corpus
>
>
> **Comments**
>
> > *CA: BART is not an autoregressive model. It is more of a seq-2-seq model.*
>
> We used the term autoregressive simply to denote that the model generated texts autoregressively. We agree with the reviewer that BART is more of a seq2seq model. We will update the terms to avoid confusion.
>
> > *CB: The captions under all the tables and figures are very brief, and they barely convey anything. For example, it is not clear in Tables and 5 and 6 it is not clear whether the reported numbers are on dev or test. Or what do the underlined numbers exactly mean. One can only guess, and that is not very helpful.*
>
> We apologize for the lack of clearness. The reported numbers in Table 5 and 6 are on the test set. The highlighted numbers denote the highest scores and the ones close to these are underlined. We will update Table 5 and 6 for better interpretability.
>
> **Regarding the paper’s weaknesses**
>
> > *Sometimes, it is hard to follow the discussion of the results, especially since the table captions are very brief.*
>
> We apologize for the confusion. We will reformat the table and update more descriptive captions for better interpretability.
>
> > *In the contributions, the authors mention text-image retrieval, which is not discussed anywhere in the main body of the paper, only in the appendix. Anything discussed in the main body of the paper should be presented there.*
>
> We agree with the reviewer. We will reorganize the sections and include text-image retrieval in the paper’s main body.
>
> > *The terminology of some know models and approaches are misleading (see comment).*
>
> We apologize for the confusion. We are happy to clarify any terminology used in the paper.
>
> **Missing References**
>
> > *There has been a lot more work on Arabic since Zaidan and Callison-Burch, 2011. For example, the MADAR corpus: https://aclanthology.org/L18-1535.pdf*
>
> We agree with the reviewer that this is a related work. We will include it in the related work section.
>
> > *Even though it is not open source (but only $25), the LDC has a corpus of Vietnamese that has multiple dialects, including the central dialect: https://catalog.ldc.upenn.edu/LDC2017S01 If the authors can not access it to explore and compare to it, I believe it is worth mentioning as an existing corpus.*
>
> We agree with the reviewer that the corpus is relatable. However, this corpus was developed for speech-related tasks (we discussed speech-related works in line 584-591 of Section 5) and did not contain parallel input pairs for central and northern dialects . In contrast, our corpus focuses on the textual attributes and their effects on downstream tasks, and also provides parallel input pairs. We will explicitly mention this corpus for better clarity.
>
> **Typos Grammar Style**
>
> > - *Any mention of Table and Figure should be capitalized.*
> > - *Line 082 Table A --> Figure 1*
> > - *Line 072 exits --> existing*
> > - *Line 103 dialect --> dialects*
>
> We agree with the reviewer on these typos and grammar errors. We will fix them.
>
>
> **Presentation Improvement**
>
> > *There is plenty of space in the paper, so it is worth making Tables 5 and 6 larger.*
>
> We agree with the reviewer. We will enlarge Table 5 and 6 for better readability.
>
> > *Adding examples from the corpus would be helpful.*
>
> We will update additional examples in the appendix section.
>
>
> **Regarding reproducibility**
>
> We will publicly release the corpus under a Creative Common license. All fine-tuned models as well as their outputs and translated responses (including those from the ChatGPT system) will be publicly accessible as well. In addition, the training and evaluation codes will be made public. We will include all details in the GitHub repository of the paper. We look forward to hearing any additional concern that you might have regarding the paper’s reproducibility.

---

### Official Review · Reviewer_64HP · 2023-08-04

**Soundness:** 3

**Excitement:**

3: Ambivalent: It has merits (e.g., it reports state-of-the-art results, the idea is nice), but there are key weaknesses (e.g., it describes incremental work), and it can significantly benefit from another round of revision. However, I won't object to accepting it if my co-reviewers champion it.

**Paper Topic And Main Contributions:**

This work aims to address the scarcity of resources for the Vietnamese central dialect. It introduces a curated parallel corpus encompassing central and northern dialects, facilitating the development of more inclusive NLP models capable of handling central-style text. Through experimentation the study shows that monolingual models achieve better results for dialect transfer tasks. It also proposes fine-tuning techniques to enhance existing NLP models' performance in the central dialect domain.

**Questions For The Authors:**

- Will the corpus and models  that you created be made available for public use?

**Reasons To Accept:**

- The work presents a new resource and models that can positively impact NLP tasks for Vietnamese Central dialect text.

**Reasons To Reject:**

- The paper is not always easy to follow and some of the terminology used is confusing.
- It is not clear whether the corpus and models developed will be availed for public use.
- The details of how the authors finetuned various models, are vague
- The paper addresses a localized issue of particular significance. However, the authors have not demonstrated how this work can be of interest to a wide audience base, even in Vietnam.

**Reproducibility:**

4: Could mostly reproduce the results, but there may be some variation because of sample variance or minor variations in their interpretation of the protocol or method.

**Reviewer Confidence:**

3: Pretty sure, but there's a chance I missed something. Although I have a good feel for this area in general, I did not carefully check the paper's details, e.g., the math, experimental design, or novelty.

**Typos Grammar Style And Presentation Improvements:**

- Page 2 , line 072 :  However, *exist* research works -> *existing*
- Page 2, line 102-103: *Albeit* a slight lexical difference between the two *dialect*, no system *manage* to correctly -> *Despite* a slight lexical difference between the two *dialects*, no system *manages* to correctly
- Page 3, since the procedure described in section 2.1 outlines what has actually been done, all the verbs in this section should be changed to past tense.
- Further to the previous comment on section 2.1, throughout the paper the authors seem to use the present tense, when in fact they should be using the past tense.
- Page 5, line 343: We partition the original dataset with... -> It is not clear here what is meant by the 'original dataset'
- Text in table 5 is too small making it very difficult to read
- Page 7, not sure why you refer to the results in table 10 before those of 9. Why not just switch the 2 tables?

---

> ### Author Rebuttal · Authors · 2023-08-28
>
> Dear reviewer,
>
> We greatly appreciate your detailed and constructive comments and suggestions. Below we address the main concerns raised in your review.
>
> **Question from the reviewer**
> > *Will the corpus and models that you created be made available for public use?*
>
> The constructed corpus will be publicly released to the community under a Creative Commons license. The fine-tuned models and their outputs as well as translated responses (including those from the ChatGPT system) will also be made available for public use. In addition, we will upload the corpus and fine-tuned models to the HuggingFace Hub [1] so that everyone can re-use them conveniently. We will include all details in the GitHub repository of the paper.
>
> [1] https://huggingface.co/
>
> **Regarding the paper’s weaknesses**
>
> > *The paper is not always easy to follow and some of the terminology used is confusing.*
>
> We apologize for the confusion. We will check the paper thoroughly and update the writing for better readability. We would also like to hear from you if there are any particular sections/terminologies that could have been expressed better so that we can clarify them to you and also make them easier to follow.
>
> > *It is not clear whether the corpus and models developed will be availed for public use.*
>
> We apologize for the lack of clearness. We resolved this issue in your above question.
>
> > *The details of how the authors finetuned various models, are vague*
>
> We mentioned details of the fine-tuning process in Section 4 (Line 344-356) and Appendix A which already included the hyperparameters (batch size, learning rate, epochs, etc.), optimizer/library choices as well as checkpoints information. In the revision, with one additional page, we will move all the information into the main paper. We would also like to hear from you if there are any particular details that need further clarity so that we can address them for you.
>
> In addition, we will make the training/evaluation codes publicly available in the GitHub repository of the paper so that these experiments can be conveniently reproduced.
>
> > *The paper addresses a localized issue of particular significance. However, the authors have not demonstrated how this work can be of interest to a wide audience base, even in Vietnam.*
>
> We study and address the dialectal bias in Vietnamese NLP models where the central-style texts are not well comprehended. As discussed in Section 1, the central region accounts for one-third of the country’s population and forms an important, idiosyncratic part of its culture. By providing remedies to the problem, we make NLP models more functionable towards a large group of audience (i.e. native users of the central dialect) and the work therefore not only benefits the research community but only promotes usability towards prospective end-users. For example, we demonstrated that the corpus could facilitate higher translation quality in the central dialect domain through fine-tuned adapters, which you also acknowledged. This will practically help the increasing number of tourists [2] who often rely on translation technologies better navigate their way through the central region in Vietnam. As a result, the work is also of importance to the development of industrial applications. Ultimately, the work should be of interest to a wider audience base, including not only Vietnamese people but also companies developing applications that are currently hindered by the dialectal bias.
>
> [2] Baum, A. (2020). Vietnam's Development Success Story and the Unfinished SDG Agenda. TransportRN: Transportation & Sustainability (Topic).
>
> **Typos Grammar Style**
> > - *Page 2 , line 072 : However, exist research works -> existing*
> > - *Page 2, line 102-103: Albeit a slight lexical difference between the two dialect, no system manage to correctly -> Despite a slight lexical difference between the two dialects, no system manages to correctly*
> > - *Page 3, since the procedure described in section 2.1 outlines what has actually been done, all the verbs in this section should be changed to past tense.*
> > - *Further to the previous comment on section 2.1, throughout the paper the authors seem to use the present tense, when in fact they should be using the past tense.*
>
> We agree with the reviewer on these grammar errors. We will fix them.
>
> **Presentation Improvement**
> > *Page 5, line 343: We partition the original dataset with... -> It is not clear here what is meant by the 'original dataset'*
>
> We apologize for the confusion. Here *'the original dataset'* literally means the constructed dataset (all samples, without separate train/val/test splits). We will change it to *'the dataset'*.
>
> > *Text in table 5 is too small making it very difficult to read*
>
> We will update the table’s size and enlarge the text.
>
> > *Page 7, not sure why you refer to the results in table 10 before those of 9. Why not just switch the 2 tables?*
>
> We apologize for the confusion. There was an error in the latex file’s reference order. We will fix it in the revision.
>
>
> ***
> \
> We hope that you can reconsider the review score. Please let us know if you would like us to do anything else.

---

### Official Review · Reviewer_rL5s · 2023-08-05

**Soundness:** 4

**Excitement:**

3: Ambivalent: It has merits (e.g., it reports state-of-the-art results, the idea is nice), but there are key weaknesses (e.g., it describes incremental work), and it can significantly benefit from another round of revision. However, I won't object to accepting it if my co-reviewers champion it.

**Paper Topic And Main Contributions:**

- The paper proposes to address the linguistic variations within the Vietnamese language, particularly focusing on its central dialect, which has received relatively less attention in existing natural language processing (NLP) research. While prior works have largely concentrated on the standard dialect and its close relationship with the southern variant, the paper acknowledges the unique characteristics of the central dialect, including distinct vocabularies and expressions, as well as its reduced mutual intelligibility with the standard dialect.

- To bridge this gap and enhance NLP capabilities for the central dialect, the paper introduces a new parallel corpus specifically designed for transferring text between the central and northern dialects. Through a series of extensive experiments, the study explores the effectiveness of different language models for dialect transfer tasks.

- The paper demonstrates that dedicated monolingual language models exhibit superior performance compared to their multilingual counterparts for handling dialect transfers. Additionally, the paper investigates the potential of fine-tuned transfer models, which act as adapters to improve the performance of existing NLP models on central dialect text. These fine-tuned models show promising outcomes in tasks such as translation and text-image retrieval.


**Reasons To Accept:**

The paper's main objective is to fill the gap in Vietnamese NLP research by focusing on the central dialect, introducing a new corpus, evaluating different language models for dialect transfer tasks, and showcasing the benefits of fine-tuned transfer models for enhancing NLP performance in the context of the central dialect.

**Reasons To Reject:**

The abstract is long, lacking conciseness and a comprehensive overview of the paper's primary points. It lacks specific summaries about the extent of the improvements and the results of the translation and text-image retrieval tasks. Removing unnecessary information and including more specific information about the experimental outcomes would provide a clearer picture of the study's findings.

**Reproducibility:**

4: Could mostly reproduce the results, but there may be some variation because of sample variance or minor variations in their interpretation of the protocol or method.

**Reviewer Confidence:**

3: Pretty sure, but there's a chance I missed something. Although I have a good feel for this area in general, I did not carefully check the paper's details, e.g., the math, experimental design, or novelty.

---

> ### Author Rebuttal · Authors · 2023-08-28
>
> Dear reviewer,
>
> We greatly appreciate your efforts in reviewing our paper and providing positive feedback on our paper. Thank you for the suggestions. We will update the abstract to remove unnecessary details and condense information to better reflect the study’s findings.

---

### Meta-Review · Area_Chair_Hb9h · 2023-09-07

**Recommendation:** 4

**Metareview:**

This paper introduces a novel parallel corpus that transfers text between the northern and central Vietnamese dialects, addressing a gap in NLP research with respect to the central Vietnamese dialect. While the potential impact of this work, the corpus creation process, and the inclusion of benchmarking experiments deserve praise, there are concerns over style and presentation. Certain sections are difficult to follow such as the Abstract and Discussion, and sometimes the terminology used is unclear. Some details are also vague or missing, e.g. how were the models fine-tuned and what were the detailed results from the text-image retrieval task. These concerns were addressed by the authors in their rebuttals, however, where they acknowledged the areas that need to be reworked and that many of the missing details can be found in the appendices (but should and would be moved into the main body of the paper).

To summarize, the revisions required to address paper content concerns can be accomplished quite readily, but substantial revisions may be required to ensure the writing is clear and up to standard.

---

### Decision · Program_Chairs · 2023-10-07

**Decision:**

Accept-Findings

**Comment:**

This paper introduces a novel parallel corpus that transfers text between the northern and central Vietnamese dialects, addressing a gap in NLP research with respect to the central Vietnamese dialect. While the potential impact of this work, the corpus creation process, and the inclusion of benchmarking experiments deserve praise, there are concerns over style and presentation. Certain sections are difficult to follow such as the Abstract and Discussion, and sometimes the terminology used is unclear. Some details are also vague or missing, e.g. how were the models fine-tuned and what were the detailed results from the text-image retrieval task. These concerns were addressed by the authors in their rebuttals, however, where they acknowledged the areas that need to be reworked and that many of the missing details can be found in the appendices (but should and would be moved into the main body of the paper).

To summarize, the revisions required to address paper content concerns can be accomplished quite readily, but substantial revisions may be required to ensure the writing is clear and up to standard.